# 3D Autonomous Surgeon’s Hand Movement Assessment Using a Cascaded Fuzzy Supervisor in Multi-Thread Video Processing

**DOI:** 10.3390/s23052623

**Published:** 2023-02-27

**Authors:** Fatemeh Rashidi Fathabadi, Janos L. Grantner, Saad A. Shebrain, Ikhlas Abdel-Qader

**Affiliations:** 1Electrical & Computer Engineering Department, Western Michigan University, Kalamazoo, MI 49008, USA; 2Department of Surgery, Homer Stryker MD School of Medicine, Western Michigan University, Kalamazoo, MI 49008, USA

**Keywords:** laparoscopic surgical skill assessment, multi-class object detection, fuzzy logic-based decision support system, intelligent box-trainer system

## Abstract

The purpose of the Fundamentals of Laparoscopic Surgery (FLS) training is to develop laparoscopic surgery skills by using simulation experiences. Several advanced training methods based on simulation have been created to enable training in a non-patient environment. Laparoscopic box trainers—cheap, portable devices—have been deployed for a while to offer training opportunities, competence evaluations, and performance reviews. However, the trainees must be under the supervision of medical experts who can evaluate their abilities, which is an expensive and time-consuming operation. Thus, a high level of surgical skill, determined by assessment, is necessary to prevent any intraoperative issues and malfunctions during a real laparoscopic procedure and during human intervention. To guarantee that the use of laparoscopic surgical training methods results in surgical skill improvement, it is necessary to measure and assess surgeons’ skills during tests. We used our intelligent box-trainer system (IBTS) as a platform for skill training. The main aim of this study was to monitor the surgeon’s hands’ movement within a predefined field of interest. To evaluate the surgeons’ hands’ movement in 3D space, an autonomous evaluation system using two cameras and multi-thread video processing is proposed. This method works by detecting laparoscopic instruments and using a cascaded fuzzy logic assessment system. It is composed of two fuzzy logic systems executing in parallel. The first level assesses the left and right-hand movements simultaneously. Its outputs are cascaded by the final fuzzy logic assessment at the second level. This algorithm is completely autonomous and removes the need for any human monitoring or intervention. The experimental work included nine physicians (surgeons and residents) from the surgery and obstetrics/gynecology (OB/GYN) residency programs at WMU Homer Stryker MD School of Medicine (WMed) with different levels of laparoscopic skills and experience. They were recruited to participate in the peg-transfer task. The participants’ performances were assessed, and the videos were recorded throughout the exercises. The results were delivered autonomously about 10 s after the experiments were concluded. In the future, we plan to increase the computing power of the IBTS to achieve real-time performance assessment.

## 1. Introduction

Laparoscopic surgery is one of the most remarkable areas of computer-integrated surgery, facilitating minimally invasive surgery (MIS) solutions to shorten recovery time and hospital stays of those surgeries with larger incisions that may lead to bleeding and infections [1,2]. However, for laparoscopic surgery, surgeons must improve their hand skills, such as incision, a variety of suturing and knotting techniques, transferring, and needle insertion before carrying out real-life operations [3].

### 1.1. Statement of the Problem

To enable training procedures in a non-patient environment, several simulation-based, advanced training systems have been developed. Laparoscopic box trainers, i.e., low-cost, portable devices, have been used for a long time to provide training experiences, skill assessments, and performance evaluations [4]. However, currently, some expert medical personnel are needed to supervise the trainee and assess his/her skills. It is a time-consuming and not-cost-effective process [5]. Simulation scenarios are conducted using a modified box trainer augmented with mathematical models to quantify the surgeon’s performance. The plain box trainer is an essential element/set of equipment that has been used to assess trainees’ fundamental laparoscopic skills to ensure whether these basic principles are mastered in a relatively similar working cavitary environment before using these laparoscopic skills on humans (abdominal cavity and chest) during laparoscopic surgery. Therefore, the simulation scenarios employed are closely related and highly similar to laparoscopic surgery. Adding an autonomous, objective performance assessment will enable the trainee to perform procedures without the need for immediate supervision by another surgeon/expert at the time trainee performs the procedure. Additionally, to prevent any intraoperative problems and malfunctions in an actual laparoscopic operation and in human intervention, a high level of surgical skill assessment is essential [6,7]. To address these problems, an autonomous skill assessment system should be developed to evaluate a trainee or a surgeon’s skills while carrying out exercises on the IBTS. One of the advantages of an autonomous skill assessment system is that even if the trainee or the supervisor missed a mistake, the system will record it and report it [8]. Using an autonomous assessment approach will open up more time for residents to practice without requiring the presence of a supervisor surgeon. This will also free up the supervisor surgeon to carry out other duties [9]. In this paper, an autonomous surgeon’s hand motion assessment system, in 3D, is proposed. It can detect and track the surgeon’s hand movements during the experiment through a multi-thread video processing implementation. 

This paper is organized as follows: In Section 1.2, recent related works regarding laparoscopic trainees’ skill assessments are reviewed. In Section 2, a detailed explanation of the methodology employed in this research is outlined. In Section 3, the experimental results are presented and discussed. In Section 4, conclusions and future plans are given.

### 1.2. Related Works

To prepare a trainee for a real operation, several hands-on skill tests in an FLS laparoscopic box trainer must be mastered. However, according to papers in surgical education, the surgeon’s skill evaluation and the capabilities of the assessment methods are fundamental matters after skill training [10]. In recent years, several researchers have studied surgeons’ performances and skill assessments by using simulators or box trainers [11,12,13,14,15,16,17,18,19,20,21,22,23,24,25,26,27]. The main focus of this section is on recent research on skill assessments by applying artificial intelligence and deep learning methods while using IBTSs and other systems. 

Grantner et al. [11] proposed an intelligent fuzzy assessment system which was able to measure the forces applied by the jaws of the grasping tools and track the tooltip movements around the test boards in a 3D virtual space. Rashidi et al. [12] developed a multi-input–single-output fuzzy logic supervisor for surgical skills assessment in a FLS pattern cutting test, based on multi-class detection and tracking of laparoscopic instruments. In this study, two state-of-the-art detectors, SSD ResNet50 V1 FPN and SSD Mobilenet V2 FPN, were trained on the new dataset, generated by the authors. By tracking the laparoscopic instruments and measuring the distance between the detected tooltip and the target object, the authors were able to evaluate the surgeon’s performance. 

Hong et al. [13] proposed a fuzzy system to consider visual supervision and assessment of the trainees’ performance in a peg-transfer task. They proposed a fuzzy system using five inputs: the distance between an instrument’s tip position and the target, the scissors’ status (whether they are open or closed), the status of an object (whether it is on a peg or not), the measured dimensions of the triangle-shaped object, and the distance between an instrument’s tip position and the center of the object. Based on these defined inputs and fuzzy rule sets, the performance of a trainee was assessed. In another research, Grantner et al. [14] proposed a tooltip movement tracking method based on a fuzzy logic system and edge detection method in a 3D virtual space. Rashidi et al. [15] detected three classes of laparoscopic instruments in a pattern-cutting task, which was performed in an IBTS using the Detecto module, built on top of PyTorch. The authors extended their work in another research project [16] and developed an intelligent box trainer performance assessment system based on the detection and tracking of multiple classes of laparoscopic instruments using the SSD-ResNet50 V1 FPN architecture. In that approach, several components were considered for the surgeon’s performance assessment, such as the surgeon not cutting and the surgeon meeting the evaluation and performance assessment criteria. A circle-cutting warning was issued if the defined criteria were not met.

Islam et al. [17] developed a web-based tool that provided assessment scores and an analysis of the surgeons’ actions over time. For this purpose, the surgeons uploaded their MIS training videos, and the system evaluated their performances based on object segmentation, motion detection, feature extraction, and error detection. Castillo-Segura et al. [18] developed an IoT learning environment to implement two FLS skills: peg-transfer tasks and precision cutting. In this method, the data were collected using sensors, and a visualization method was applied to provide feedback for surgical skills assessment. Fekri et al. [19], proposed an assessment system for analyzing the surgeons’ improvements in surgical drilling operations based on a deep recurrent neural network with a long short term memory (LSTM) architecture that models an expert person’s behavior. The network was trained using data obtained from the drill’s temperature, penetration depth sensors, and the type of bone layer to assess the performance.

Mohaidat et al. [20] proposed an autonomous skill assessment system to monitor intracorporeal suturing, one of the critical hands-on tests in FLS training, by using various versions of one-stage object detectors such as YOLOv4, Scaled-YOLOv4, YOLOR, and YOLOX. The authors proposed another skill assessment system in [21]. This study used an object detection algorithm, Scaled-YOLOv4, in conjunction with a centroid tracking algorithm, to evaluate the surgeon’s skills during advanced intracorporeal suturing. They proposed a system capable of locating and tracking surgical instruments and providing an evaluation of the performance of a surgeon.

Alkhamaiseh et al. [22] created a new dataset for one of the box-trainer pattern cutting tests, in which two circles were drawn on an artificial tissue. The radius of the inner circle and the outer circle were 2.5 and 3.0 cm, respectively. The authors proposed a method for detecting laparoscopic surgical instruments and other objects for that specific task using YOLOv5 and scaled-YOLOv4 based on the cross-stage partial (CSP) network and validated the accuracy of the model using mean average perdition (mAP), recall, and precision. 

Jago et al. [23], used quantitative variables such as completion time, symmetry ratios (non-dominant vs. dominant tool motion), and surgical tool translation (sagittal, frontal, and coronal planes; surgical tool path-length), derived from 3D motion analysis, to assess laparoscopic surgical skill level. In this experiment, all participants watched an instructional video before data collection and carried out the task three times. After that, in order to evaluate each surgeon’s skill level with a box trainer, a 3D motion capture system recorded trajectories of retroreflective markers placed on two graspers and computed the locations of the surgical tool tips. However, the proposed evaluation models needed improvement by adding feedback, with respect to the quality of the movements. In order to investigate the usefulness of simulator usage for novice surgeons, a surgical simulator for spreader graft placement was used by C. J. Oh et al. [24]. In this experiment, two groups of residents and experts were instructed to secure the two spreader graft specimens into a defined position. To assess the surgeon’s skill level and the advantage of the simulator usage, the time which was required to finish the suturing task, the cumulative number of hand motion direction changes, the total hand displacement, and the accuracy of the suture insertion were measured. In this study, hand motion was tracked by using an electromagnetic position-sensing device.

Gao et al. [25] evaluated the learning curve data over three weeks in order to train a multivariate, supervised machine learning model known as the kernel partial least squares, in a pattern-cutting task on an FLS box trainer and a virtual basic laparoscopic skill trainer (VBLaST). An unsupervised machine learning model was utilized for the trainees’ skill level classification. Using machine learning algorithms, the authors assessed the performances of trainees by considering their performances during the first 10 repetitive trials and predicted the number of trials required to achieve proficiency and the final performance level. However, there was a lack of information regarding how the dataset was prepared and what the evaluation criteria were, and there was no visualization of the performance assessment. Kuo et al. [26] developed a method for skill assessment of the peg-transfer task in a laparoscopic simulation trainer based on machine learning algorithms to distinguish between experts and novices. 

One of the main objectives of this study was to assess surgeons’ hand movements in a 3D space, in an autonomous fashion. Although in our previous works, we improved autonomous skill assessment systems for the peg transfer and tooltip tracking tasks, they were not implemented in a 3D-space environment. In previous works, we used only one camera or three cameras, working separately; hence, we were able to track in 2D space. In this study, the autonomous assessment system can detect the laparoscopic instruments and track the surgeon’s hand movements during the experiment by applying a multi-threaded video processing method which can perform 3D supervision and evaluation. In this work, two types of datasets, with time series data and image data, were used. The movement of the eyes was recorded as a time series of the gaze-point coordinates, and it was the input to a computer program to find a classification of the surgeons’ skills. Additionally, images were extracted from videos, where they were labeled according to the corresponding skill levels, and they were fed into the Inception-ResNet-v2 to classify the image data. Finally, the outcomes from the time series and the image data were used as inputs to extreme learning machines (ELM) to assess the trainee’s performance. Table 1 presents a summary of the related works and the contribution of the current study in comparison with the previous works:

## 2. Materials and Methods

This study and methodology were approved by doctors at the Homer Stryker M.D. School of Medicine, WMU (WMed). Nine WMed physicians (surgeons, residents from the surgery and OB/GYN residency programs) took part in this experiment for about three years. Three residents were categorized as laparoscopic novices (no experience in laparoscopic surgery) and six doctors as laparoscopic experts (having completed several laparoscopic procedures) and they were all right-handed. All participants were oriented and instructed to use the box trainer and the laparoscopic training simulator and given a demonstration of both. The experiment was the peg-transfer task, in which a trainee grasps each triangle object with the non-dominant hand and transfers it in mid-air to the dominant hand. Then the triangle object is put on a peg on the opposite side of the pegboard, Figure 1. There is no requirement on the color or order in which the six triangle objects are transferred. Once all six objects have been transferred to the right side of the board, the trainee should transfer them back to the left side of the board [27]. 

To assess surgeons’ hand movement in the IBTS in 3D, an autonomous assessment system is proposed which can detect the laparoscopic instruments and track the surgeon’s hand movements during the experiment by applying a multi-threaded video processing method. To have 3D tracking capabilities, the x and y dimensions are taken from the top camera, as illustrated in Figure 2a, in which the top view has been divided into three regions: Close, Middle, and Far. The z dimension is taken from the front camera, and the front view was divided into three regions, too: High, Field, and Down, as illustrated in Figure 2b. This method was implemented by the Tensorflow Object Detection API and a cascaded fuzzy logic assessment system. The latter is made up of two fuzzy logic systems working in parallel in the first level to assess the left and right hands’ movements simultaneously. These two systems are cascaded with the final fuzzy logic assessment system in the second level. A deep neural network model, explained in [16], is deployed to detect the laparoscopic instruments by capturing video images from the top and front cameras. The cascaded fuzzy logic system delivers the assessment of the surgeon’s hand movements to the left and right and up and down. 

To simplify the variable definition, a nomenclature is presented in Table 2, including all the parameters which were used in the design of the cascaded fuzzy logic supervisor. 

### 2.1. Dataset Preparation and Model Training

The experimental peg-transfer task was carried out by nine physicians (surgeon, surgical, and OB/GYN residents) in the Intelligent Fuzzy Controllers Laboratory, WMU, as illustrated in Figure 3. They possessed different levels of skill. To create our first laparoscopic box-trainer dataset (IFCL.LBT100), 55 videos of peg-transfer tasks were recorded using the top, side, and front cameras of our IBTS in the Intelligent Fuzzy Controllers Laboratory, at WMU. More than 5000 images were extracted from those videos and were labeled manually by the drawing of bounding boxes around the objects by an expert. Of all the images, 80% of them were used for training and 20% for evaluation purposes. In addition, we wanted to prepare a dataset using the residents’ recorded performances in the peg-transfer task to apply a machine learning algorithm that would predict if a resident is ready for a real laparoscopic surgery procedure. To support research in the Laparoscopic Surgery Skill Assessment area, a dataset has been created and made available, free of charge, at: https://drive.google.com/drive/folders/1F97CvN3GnLj-rqg1tk2rHu8x0J740DpC (accessed on 12 January 2022).

In this study, as in our previous works [16,28], the SSD ResNet50 V1 FPN Architecture was used as a feature extractor, and the created dataset was trained on this model. The SSD with ResNet50 V1 FPN feature extractor (Figure 4) predicts the existence of an object in different scales based on several common features, such as color, edge, centroid, and texture [12]. 

Object tracking is a process of object feature detection in which unique features of an object are detected. In this work, to detect and track the object based on the ResNet50 V1 FPN feature extractor, the related bounding box corresponding to a ground-truth box should be found. All boxes are on different scales, and the best Jaccard overlap [29] should be found over a threshold of 0.5 to simplify the learning problem. A momentum optimizer with a learning rate of 0.04, which is not very small—neither is it a large one—was used for the region proposal and classification network. Obviously, a very small learning rate slows down the learning process, but it leads to smooth convergence, whereas a large learning rate speeds up the learning but may not allow convergence [30]. 

The input to the network is an image with the arbitrary size of 1280 × 720 pixels, and the frame rate is of 30 frames per second (FPS). The final output is a bounding box for each detected object as a class label, along with its score. To carry out the experiments, the IBTS software was developed using the Python programming language in the Tensorflow deep learning platform [31]. The feasibility of this work has been evaluated using the TensorFlow API running on Windows, with a NVIDIA GeForce GTX 1660 GPU. After training the model using the customized dataset, model validation was carried out on the test set as the input images of the trained model. A loss function was used to optimize the model, and it is defined as the difference between the predicted value of the trained model and the true value of the dataset. The loss function used in this deep neural network is cross-entropy. It is defined in Equation (1) [30,32]: (1)$$\mathrm{cross}-\mathrm{entropy}=-\overset{n}{\underset{i=1}{\sum }}\overset{m}{\underset{j=1}{\sum }}y_{i,j}\log\left(p_{i,j}\right)$$
where $y_{i,j}$ denotes the true value, for example, 1 if sample *i* belongs to any of those 10 j classes, and 0 if it doesn’t belong to any of them; and $p_{i,j}$ denotes the probability, predicted by the trained model, that sample i belongs to any of those 10 j classes. The train-validation total loss at the end of 25,000 epochs for the SSD ResNet50 v1 FPN was about 0.05. Many object detection algorithms use the mean average precision (mAP) metric to analyze and evaluate the performance of object detection in trained models. The mean of average precision (AP) values, which can be based on the intersection over union (IoU), are calculated over recall values from 0 to 1. IoU designates the overlap of the predicted bounding box coordinates to the ground-truth box. Higher IoU means the predicted bounding box coordinates closely resemble the ground-truth box’s coordinates [33]. The IoU in this study is 0.75; therefore, based on [34], the performance of this learning system is high enough. Additionally, the mAP has been used to measure the accuracy of the object detection model, which reached 0.741. 

### 2.2. Distance Measurement Method in Threaded Video Processing

In this method, the surgeon’s hand movements are assessed by tracking the laparoscopic instruments. Using the Tensorflow Object Detection API, each object is detected and tracked during the IBTS skill training. The trained model runs inferences that return a dictionary with a list of bounding boxes’ coordinates [xmax, xmin, ymax, ymin] with prediction scores. To implement the tracking procedure, the tracking point must be located frame-by-frame in laparoscopic videos. The output of the instrument/object detection models comprises the predicted instruments for every frame, along with the x and y coordinates of their associated bounding boxes. After running the Tensorflow Object Detection API on the trained model, boxes and labels of each object are visualized in each frame for both threaded cameras, simultaneously. The coordinates of each bounding box of each object are used for the centroid calculation [35]. To measure the actual coordinates and the centroids of the bounding boxes in centimeters, the frame width, $x_{f}$, and frame height, $y_{f}$, based on pixels, should be obtained. Using these measurements, the actual frame width and height of the frame (based on centimeters) are measured. Using the following calculation (Equaiton (2)), the average size of a pixel per centimeter is calculated. The top-left corner of the frame is the origin of the coordinates:(2)$${\;\mathrm{average}}_{{\mathrm{px}}_{{\mathrm{cm}}_{w}}}=\frac{\mathrm{frame}\;\mathrm{width}\left(\mathrm{pixel}\right)}{\mathrm{real}\;\mathrm{frame}\;\mathrm{width}/\mathrm{height}\left(\mathrm{cm}\right)}$$

The centroid of the right and left graspers in the top and front cameras is an approximate prediction of each grasper tip’s location, which is used to generate the inputs of the fuzzy logic systems. As mentioned before, the main aim of this study was to monitor if the surgeon’s hands moved outside of the defined field (the yellow box on the surface of the IBTS in Figure 2) on the left and right sides or if they moved above the desired height from the test board. To implement this method, both cameras have to be activated at the same time and after the laparoscopic instruments are detected. The location of each instrument is calculated using the top and front cameras. This algorithm removes the need for any human monitoring and intervention. In the first level of the cascaded fuzzy logic system, where two multi-input–single-output (MISO) fuzzy logic assessments are executed in parallel, the instrument’s distance from the coordinates of the field of interest ($x_{f}$, $y_{f}$), is calculated from the first camera to obtain the first input of these two MISO parallel fuzzy logic systems, Dis_Cam1_R and Dis_Cam1_L. Using the Euclidean distance formula, Equation (3), the distances from the centroids of the right and left instrument, $\left(x_{c_{r}},y_{c_{r}}),\;(x_{c_{l}},y_{c_{l}}\right)$, to the centroid of the of the field of interest ($x_{f}$, $y_{f}$), are calculated, as illustrated in Figure 2 and Figure 5. In Equation (3), *p* and *q* are the two points in the Euclidean n-space; *q_i_* and *p_i_* are the Euclidean vectors, starting from the origin of the space (initial point). Additionally, ${(}_{\ldots },h_{c_{r}})$ and ${(}_{\ldots },h_{c_{l}})$, the second elements of the right and left graspers’ centroids, detected by the front camera, are the second input of these two MISO parallel fuzzy logic system, H_Cam2_R and H_Cam2_L; see Figure 2 and Figure 5.
(3)$$d\left(p,q\right)=\sqrt{\overset{n}{\underset{i=1}{\sum }}{\left(q_{i}-p_{i}\right)}^{2}}$$

The pseudocode for the proposed tracking and measurement algorithm (TMA) is as follows (Algorithms 1):
**Algorithms 1:** TMA**Procedure (TMA) {**Initialize camera dictionary to {‘top’: **None**, ‘front’: **None**, ‘Number’: **0**} **while** *camera counter is less than two* **do**print the camera name print camera counter **while** *True* **do**    take the trained model     take the image from the camera     run inference for single image based in the model and image to detect objects     **average_px_meter** ← calculate pixel per centimeters             calculate distances of two centroids (centroid_1, centroid_2) **calculate_centr_distances ←*math.sqrt***((centroid_2[0]-centroid_1[0])^2^ + (centroid_2[1]-centroid_1[1])^2^)     **if**
*camera name is “Top Camera”* **then {**
     **calculate** centroids[0] ← (centroids[0]/average_px_cm_h)      **calculate** centroids[1] ← (centroids[1]/average_px_cm_h)   **calculate** distance from center of field to center of right grasper ← calculate_centr_distances(center, centroids[0])      **calculate** distance from center of field to center of left grasper ← calculate centr_distances(center, centroids[1])      **add** (distance_from_center_Right, distance_from_center_Left) to camera_dic[‘top’]        **}**
    **else if** *camera name is “Front Camera”* **then** {      **calculate** height_right ← (centroids[0][1]/average_px_cm_h)      **calculate** height_left ← (centroids[1][1]/average_px_cm_h)      **add** (height_right, height_left) to camera_dic[‘front’]        **}**
  **}**

### 2.3. Cascaded Fuzzy Logic Supervisor

For autonomous, objective performance assessments for surgeons, executing a standard test is a challenging problem. There are no exact mathematical models available to quantify the performance of the surgeon, e.g., eye-hand coordination skills and trajectories of the tool movements in a limited 3D space. Measurements of significant variables help to render a decision on the performance, but ultimately, it is up to an expert surgeon to interpret the results. That includes the granularity (i.e., how many values or labels) are used in the domains of the measured variables and what kinds of overlapping value intervals are associated with those labels. These decisions are made by a domain expert. Fuzzy logic provides a mathematical framework to treat measured data, along with expert opinion. Creating a knowledge base, an inference engine, and the selection of the fuzzification and defuzzification algorithms are part of the development of a fuzzy logic-based decision support system. The application area, e.g., human performance assessment versus non-linear process control, has a significant impact on those choices. The block diagram of the system is illustrated in Figure 5.

Due to the symmetry of the background, getting far away from the center of the field of interest to the left or right side of the video frame leads to poor performance of the surgeon’s, and any action carried out around the coordinates of the center of the field is accepted as long as the calculated height by the front camera meets the range of the desired values. In other words, the right and left graspers should move around the field from the perspective of the front camera; see Figure 2. Thus, due to this symmetric calculation Equation (4), both the left-hand and right-hand tooltip movements are assessed and fuzzified in the same way; hence, the membership function (MF) and linguistic variables of the first-level of the two-MISO, parallel fuzzy logic system’s inputs for right- and left-hand movement evaluation systems are the same, as illustrated in Figure 6:
(4)$$\begin{array}{ll} \mathrm{Input}1: & {\mathrm{Dis}}_{\mathrm{Cam}1_{R/l}}\;=\sqrt{{\left(x_{c_{r/l}}-x_{f}\right)}^{2}+{\left(y_{c_{r/l}}-y_{f}\right)}^{2}}\\ \mathrm{Input}2: & H_{\mathrm{Cam}2_{R/l}}={(}_{\ldots },h_{c_{r/l}}) \end{array}$$

In this study, the Mamdani fuzzy inference engine was used, the input fuzzifier unit implemented the Singleton fuzzifier method, and the defuzzifier unit executed the center of the average algorithm [4]. The centroid defuzzification method defines the center of the fuzzy set area within the defined boundaries of each input and output (*x* axis) and returns the corresponding crisp value. The centroid is calculated using Equation (5), where *μ*(*x_i_*) is the MF value for xi in the universe of discourse.
(5)$$x_{\mathrm{centroid}}=\frac{\sum \mu \left(x_{i}\right)x_{i}}{\sum \mu \left(x_{i}\right)}$$

In consultations with WMed laparoscopic surgeons, a fuzzy IF-THEN ruleset was developed for the separate assessment of the movements of each hand. This ruleset was made up of 9 rules, as given in Table 3. These rules were set to be the same for both grasper tip movements. The output labels refer to typical letter grades. The outputs of the first-level fuzzy logic evaluation systems for the right- and left-hand movements are denoted as SRHPA and SLHPA, respectively. The output linguistic variables were mapped into five levels, A, B, C, D, and E, over the universe of discourse 0–100%, as shown in Figure 7.

In the second level of the fuzzy logic evaluation system, the performances of the surgeon’s right and left hands were also evaluated in a summary fashion. For that purpose, the SRHPA and SLHPA outputs of the first level were refuzzified, and their granulations were revised (SRHPAr and SLHPAr, respectively) to create proper fuzzy inputs for the second level over the universe of discourse of 0–100%. For the second level, 9 fuzzy IF-THEN rules were proposed as well, as indicated in Table 4. The output linguistic labels were the same as in the final surgeon’s performance assessment in the second level, and were mapped into five levels, A, B, C, D, and E [10], over the universe of discourse of 0–100%, as shown in Figure 8.

## 3. Experimental Results

The novelty of this work is the development of this 3D autonomous assessment system by monitoring the surgeon’s hands’ movements during the training exercises and assessing the performance, rather than performing that by observation. To validate the accuracy of this method, several experiments have been executed on both recorded videos and by using the IBTS in the Intelligent Fuzzy Controllers Laboratory, WMU. Our intelligent box-trainer system (IBTS) is depicted in Figure 9. Our goal was to establish an objective laparoscopic surgery skills assessment system; hence, this platform serves as our development environment for hardware, software, and algorithms. The following list includes the IBTS’s principal elements: A tablet used by medical staff, a PC workstation to record the test videos and run the tracking and assessment programs, and a router to implement wireless communications between the tablet and the PC make up the FLS box-trainer device. Additional LED strips have been used for better lighting.

### Simulation Results

To validate the proposed method for the 3D assessment of the surgeon’s hand movements during the IBTS tests using the cascaded fuzzy supervisor in multi-thread video processing, several experiments were run (Figure 10, Figure 11 and Figure 12). In the first two figures (Figure 10 and Figure 11), two specific moments in the experiments are shown. Figure 12 illustrates the experiment during running the program on two recorded videos. In what follows, a detailed explanation of each experiment is presented. In these experiments, the top and front cameras were operating simultaneously, and after running the Tensorflow Object Detection API on the trained model, the boxes and labels of each object were visualized in each frame for both threaded cameras simultaneously. In Figure 10, one can see that the right and left graspers were detected and localized based on the proposed algorithm. As shown by the top camera, in Figure 10a, the right grasper was very close to the center of the field of interest, and the left grasper was very close to the sides of the field of interest. Based on Figure 10b, by the front camera, ${(}_{\ldots },h_{c_{r}})$ and ${(}_{\ldots },h_{c_{l}})$ were in the fields of interest, as illustrated in Figure 2. Therefore, based on the defined fuzzy rule set, the assessments for the right and left graspers were calculated as shown in Figure 10c,d. The final assessment in Figure 10e illustrates good performance.

In Figure 11, the right and left graspers are detected and localized, based on the proposed algorithm. As shown by the top camera, as depicted in Figure 11a, the right and left graspers are very close to the center of the field of interest but on the other hand, based on Figure 11b, the front camera, ${(}_{\ldots },h_{c_{r}})$ is in the field of interest area while the left grasper is not detected. In Figure 11a, we notice that the size of the left grasper is bigger than its usual size, meaning that it is very close to the top camera, and that is the reason why it is not shown in the image by the front camera Hence, the left grasper ${(}_{\ldots },h_{c_{l}})$ is in the High region. Therefore, based on the defined fuzzy rule list, the assessments for the right and left graspers are calculated, as shown in Figure 11c,d. The final assessment, as depicted in Figure 11e, illustrates only partially accepted performance.

Figure 12a–c illustrates the simultaneous presentation of the deep learning method along with the fuzzy logic system, in the multi-thread video processing implementation. As shown in these figures, the right grasper, the left grasper, and the final assessment are shown next to the threaded videos to give an illustration of the step-by-step assessment procedure, either in a real-time situation or on a recorded video. Figure 12a shows that the left grasper’s position is in an acceptable condition and the right grasper’s position is rather far from the center of the field of interest by the top camera but it has an acceptable height from the view by the front camera. The conclusion is that the surgeon is moving the object to the center of the field. A perfect performance of the surgeon’s hand movements during a transfer action is illustrated in Figure 12b. As shown in Figure 12c, it is another example when the left grasper is not detected by the top camera since it is in a high position, meaning a poor left hand movement by the surgeon.

## 4. Discussion

One of the advantages of this 3D autonomous skill assessment is that even if a trainee or a supervisor, while evaluating a task, did not notice any mistakes or marginal work, the system would declare poor performance autonomously. Moreover, the proposed 3D autonomous assessment approach will give the trainees more time to practice without needing a supervisor surgeon’s presence. This method has been executed on both recorded videos and IBTS exercises. In the first step, after implementing the algorithm on the IBTS, the recorded videos were fed to the algorithm for evaluation. In the next phase of this study, the algorithm was executed while the trainee was carrying out the peg-transfer task. Based on this experiment, the proposed algorithm had a better performance on the recorded assessment videos compared with the real-time experiments. The main reason for that is the program execution delay generated by the whole system. This delay was about 10 s in detection, because of the relatively small size of the available memory and not having a better GPU. We plan to replace the PC in the IBTS with a more powerful computer, extend its memory to 32 GB, and install the NVIDIA GeForce RTX 3080 10GB GDDR6X 320-bit PCI-e Graphics Card in the PC to reduce this delay. We hope that with these improvements, real-time performance assessment can be achieved.

## 5. Conclusions

In contrast to potential laparoscopic surgery assessment problems created by mistakes made by some supervisory personnel (e.g., misidentifying a situation), an autonomous assessment system can facilitate and accelerate the trainees’ performance assessments using a box-trainer system. One of the contributions of this study is related to multi-thread video processing, where the top and front cameras are engaged in the evaluation process, providing a 3D-space environment; thus, the chance of object detection or any malfunction recognition from three dimensions will increase. To assess the surgeons’ hand movement in the IBTS in three dimensions, a fully autonomous assessment system was proposed and implemented which has the ability to detect and track the surgeon’s hand movements during the execution of a test through a multi-threaded video processing application. This method works based on the Tensorflow Object Detection API and a cascaded fuzzy logic assessment system. A deep neural network model was deployed to detect the laparoscopic instruments by the top and front cameras. Finally, the cascaded fuzzy logic system was used to assess a surgeon’s hand movements to the left and right sides and up and down, with respect to the platform board. Based on several experiments, the proposed algorithm delivered the results faster in the case of recorded assessment videos compared with real-time experiments because of the 10 s delay in object detection. This delay was due to the less-than-required computing power of the current IBTS configuration. In the future, we plan to replace the PC in the IBTS with a more powerful computer, extend its memory to 32 GB, and install a NVIDIA GeForce RTX 3080 10GB GDDR6X 320-bit PCI-e Graphics Card in the PC to reduce this delay. Although this algorithm removed the need for a supervisor, executing this algorithm on different FLS tests on the IBTS is still a challenging problem. Hence, we plan to train a deep neural network model which can function on several FLS tests. 

## Figures and Tables

**Figure 1 sensors-23-02623-f001:**
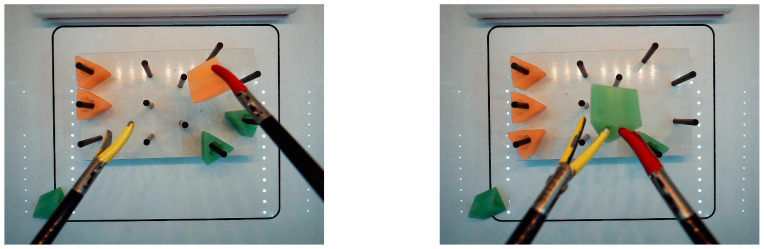
Pegboard with 12 pegs, and six ring-like objects (triangles) in the IBTS.

**Figure 2 sensors-23-02623-f002:**
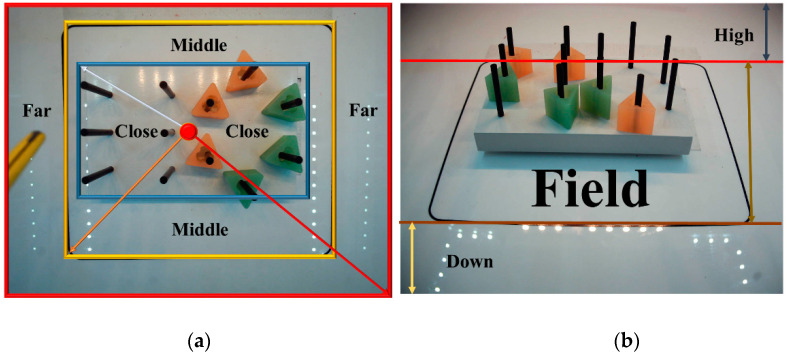
Top (**a**) and front (**b**) view segmentation for the purpose of three-dimensional assessment.

**Figure 3 sensors-23-02623-f003:**
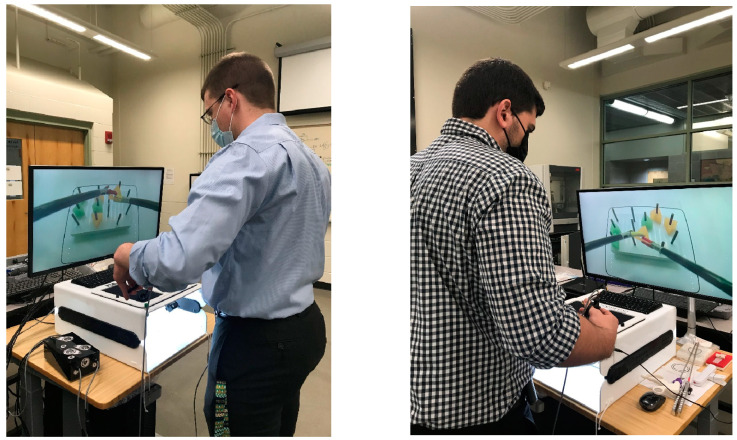
WMed surgical residents using the IBTS for laparoscopic training tasks.

**Figure 4 sensors-23-02623-f004:**
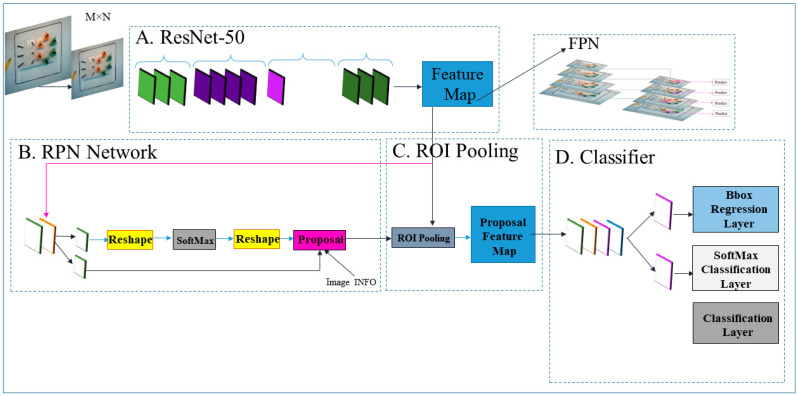
(**A**). ResNet-50. (**B**) RPN network for generating regional proposals. In this proposed detector, the RPN connects the last conv feature map generated by FPN to a sliding window. It localizes any object, along with the RPN classifies scores and bounding boxes (Bbox) of the proposed regions. In part (**C**), the last conv feature map connects to a RoI pooling layer, leading to the proposed region. Finally, in (**D**) the classifier, there are two output layers of Fast R-CNN representing two vectors per proposed region: SoftMax probabilities and Bbox regression.

**Figure 5 sensors-23-02623-f005:**
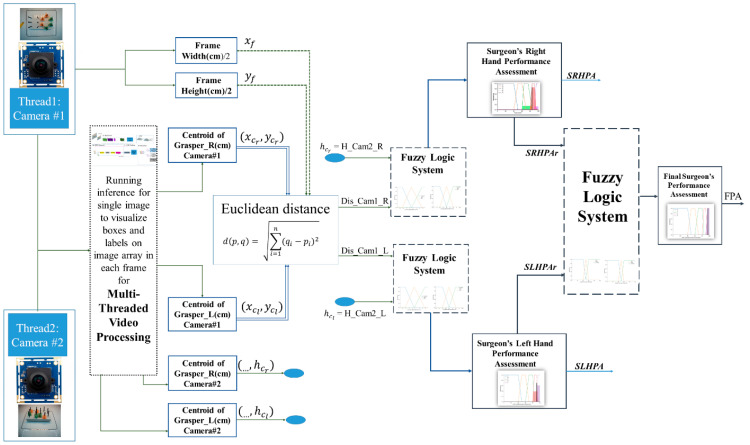
Block diagram of autonomous cascaded fuzzy supervisor assessment system in a multi-thread video processing experiment in the IBTS.

**Figure 6 sensors-23-02623-f006:**
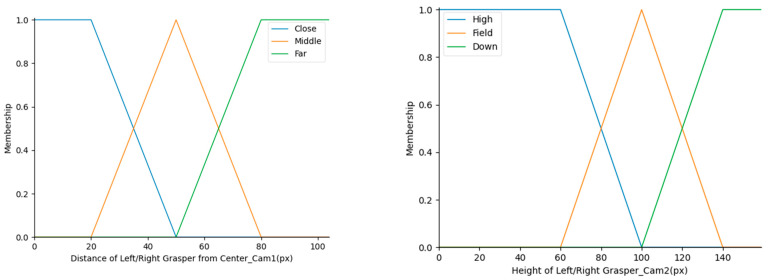
MFs of the input variable for the first level fuzzy logic evaluation system for both the **right** and **left** graspers.

**Figure 7 sensors-23-02623-f007:**
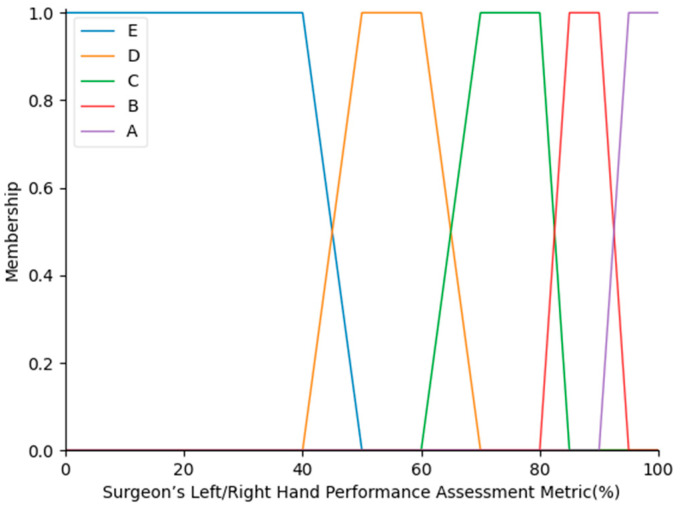
Output variable MFs for surgeon’s performance assessment for the right and left-hand movements in the first level.

**Figure 8 sensors-23-02623-f008:**
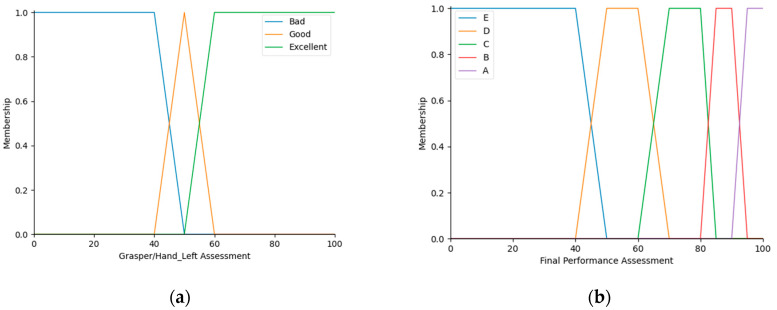
(**a**) MFs of input variables SRHPAr and SLHPAr, in the second level of the fuzzy logic evaluation system. (**b**) Membership functions of the output variable FPA in the second level.

**Figure 9 sensors-23-02623-f009:**
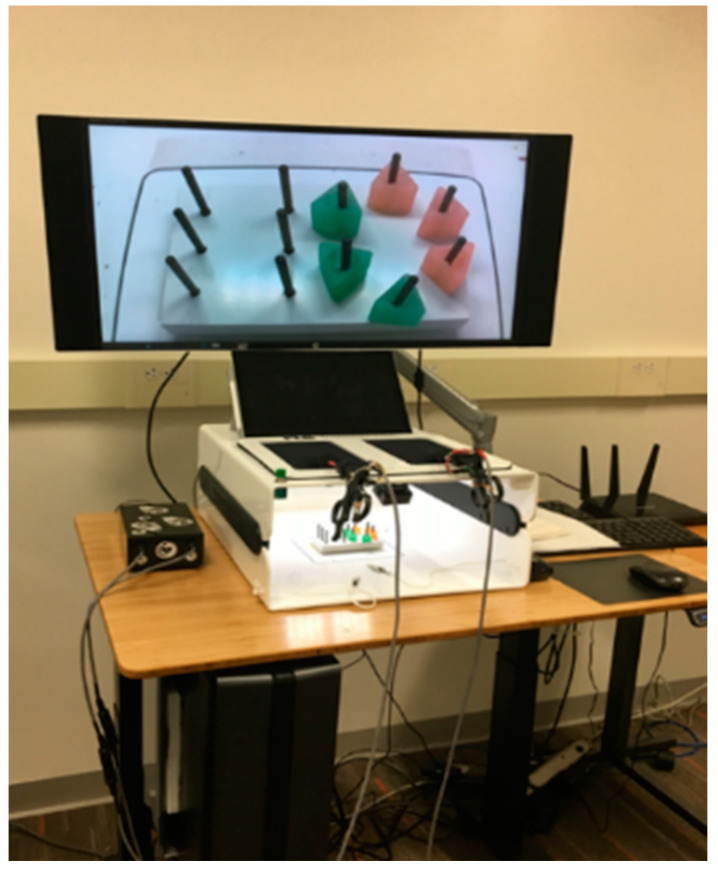
The IBTS.

**Figure 10 sensors-23-02623-f010:**
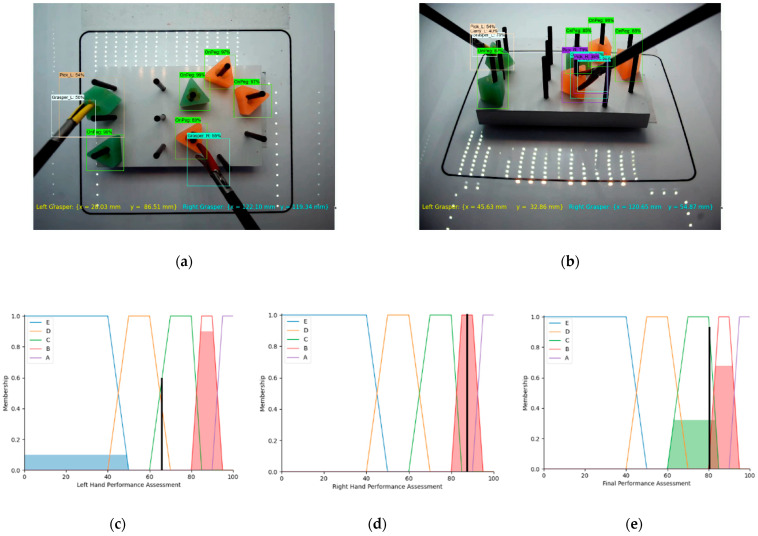
(**a**,**b**) Object detection and measurement metric results in top and front cameras, (**c**) SLHPA, (**d**) SRHPA, (**e**) final assessment FPA (good performance).

**Figure 11 sensors-23-02623-f011:**
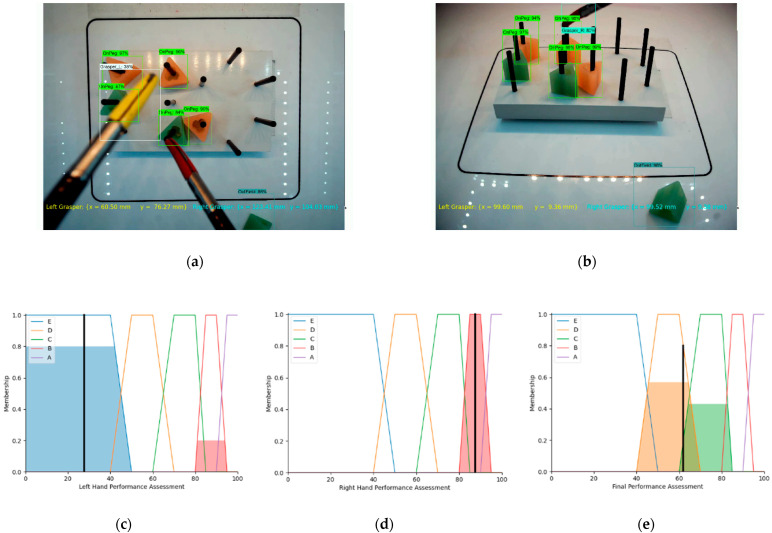
(**a**,**b**) Object detection and measurement metric results for top and front cameras, (**c**) SLHPA, (**d**) SRHPA, (**e**) final assessment FPA (not very good performance).

**Figure 12 sensors-23-02623-f012:**
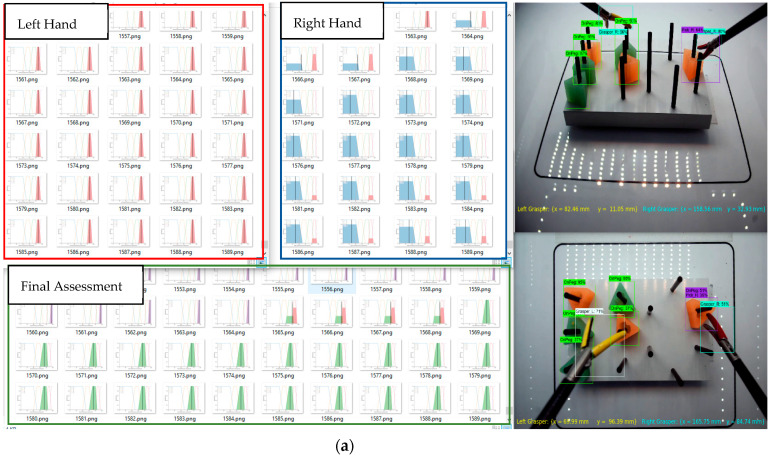
Object detection and measurement metric results by the top and front cameras, SLHPA, SRHPA, and final assessment FPA over a period of time.

**Table 1 sensors-23-02623-t001:** Summary of the related works and the contribution of the current study.

Related Studies	Related Studies’ Contribution	Remaining Issues and Challenges	Contribution
[11]	proposed an intelligent fuzzy assessment to measure the forces applied by the jaws of the grasping tools and track the tooltip movements around the test boards.	Laparoscopic box trainer dataset for deep learning methods	Solved
[12,14,15,16]	developed a performance assessment system based on the detection and tracking of a multi-class of laparoscopic instruments using the SSD-ResNet50 V1 FPN architecture and MISO logic supervisors for surgical skills assessment in a FLS pattern cutting test.	Autonomous assessment system to detect and track hand movements in IBTS	Solved
[13]	proposed a fuzzy system to consider visual supervision and assessment of the trainees’ performance of a peg transfer task.	3D object detection and tracking in a multi-thread video processing in IBTS	Solved
[17]	developed a web-based tool that provided assessment scores and an analysis of the surgeons’ actions over time.	Real-time skilled assessment	Partially Solved
[18]	developed an IoT learning environment to implement the peg transfer tasks and precision cutting.		
[19]	proposed an assessment system for analyzing the surgeons’ improvements in surgical drilling operations based on a deep RNN with a LSTMS architecture that models an expert person’s behavior.		
[20,21]	proposed an autonomous skill assessment system to monitor intracorporeal suturing, one of the critical hands-on tests in FLS training by using various versions of One-Stage-Object-Detectors such as YOLOv4, Scaled-YOLOv4, YOLOR, and YOLOX.		
[22]	proposed a method for detecting laparoscopic surgical instruments and other objects for box trainer pattern cutting task using YOLOv5 and scaled-YOLOv4, based on CSP Network.		
[23]	used quantitative variables such as completion time, symmetry ratios (non-dominant vs. dominant tool motion), and surgical tool translation (sagittal, frontal, coronal planes, surgical tool path-length), derived from 3D motion analysis to assess laparoscopic surgical skill level.		
[24]	to investigate the beneficiary of simulator usage for novice surgeons, a surgical simulator for spreader graft was used.		
[25]	evaluated the learning curve data over three weeks in order to train a multivariate, supervised machine learning model known as the kernel partial least squares, in a pattern-cutting task on an FLS box trainer and a virtual basic laparoscopic skill trainer (VBLaST).		
[26]	developed a method for skill assessment of the peg transfer task in a laparoscopic simulation trainer based on machine learning algorithms to distinguish between experts and novices.		

**Table 2 sensors-23-02623-t002:** Rule list—first-level fuzzy logic evaluation.

$\left(x_{c_{r}},y_{c_{r}}\right)$	Distance from the centroid of the right instrument
$\left(x_{c_{l}},y_{c_{l}}\right)$	Distance from the centroid of the left instrument
($x_{f}$, $y_{f}$)	Centroid of the field of interest
$x_{f}$	Frame width, based on pixels
$y_{f}$	Frame height, based on pixels
SRHPA	Surgeon’s Right Hand Assessment
SLHPA	Surgeon’s Left Hand Assessment
SRHPAr	The first input of the second-level fuzzy logic system. Refuzzified output of SLHPA from the first level.
SLHPAr	The first input of the second-level fuzzy logic system. Refuzzified output of SRHPA from the first level.
Dis_Cam1_R	The first input of the first-level fuzzy logic system, distance from the centroid of the right instrument, $\left(x_{c_{r}},y_{c_{r}}\right)$ to the centroid of the field of interest ($x_{f}$, $y_{f}$)
Dis_Cam1_L	The first input of the first-level fuzzy logic system, distance from the centroid of the right instrument, $\left(x_{c_{l}},y_{c_{l}}\right)$ to the centroid of the field of interest ($x_{f}$, $y_{f}$)
H_Cam2_R	The second input of the first-level fuzzy logic system, ${(}_{\ldots },h_{c_{r}})$, right graspers’ centroids, detected by the front camera.
H_Cam2_L	The second input of the first-level fuzzy logic system ${(}_{\ldots },h_{c_{l}})$, Left graspers’ centroids, detected by the front camera.

**Table 3 sensors-23-02623-t003:** Rule list—First-level fuzzy logic evaluation.

	$Dis_{Cam1_{R/l}}$	Close	Middle	Far
$H_{Cam2_{R/l}}$				
High		B	E	E
Field		A	B	C
Down		B	B	D

**Table 4 sensors-23-02623-t004:** Rule set—second-level fuzzy logic evaluation.

	SLHPAr	Excellent	Good	Bad
SRHPAr				
Excellent		A	B	C
Good		B	C	E
Bad		C	D	D

## Data Availability

To support research in the Laparoscopic Surgery Skill Assessment area, a dataset has been created and made available, at: https://drive.google.com/drive/folders/1F97CvN3GnLj-rqg1tk2rHu8x0J740DpC (accessed on 12 January 2022). As our research progresses, more files will be added to this dataset. The code and the dataset will also be shared on the first author’s GitHub page (after the publication of the paper at: https://github.com/Atena-Rashidi) (accessed on 12 January 2023).

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
