# Peer review of "3D Autonomous Surgeon’s Hand Movement Assessment Using a Cascaded Fuzzy Supervisor in Multi-Thread Video Processing"

_sensors, 2023, doi:10.3390/s23052623_

Round 1

Reviewer 1 Report

General remarks:

The paper is about Autonomous system to support learning and assessment of performing laparoscopic procedures.

The authors already claim in the title that it is a 3D system, although in my opinion it is not quite true - in fact, two simultaneous 2D streams are analyzed, which does not detract from the usefulness and originality of the presented solution. However, at no stage of the algorithm's operation, are the 3D data analyzed.

The beginning of the paper is quite clear, although the literature review can be a bit overwhelming. Unfortunately, the subsequent chapters are written in a manner that is less and less careful.

The whole section 2.2 should be improved, the authors should rethink the designations of the variables. It would be advisable to provide an illustration of them. At this point I feel that a very simple problem has been described in a very vague way.

I am also not convinced by the validation of the correctness of the final grades for the tasks performed, have they somehow been verified by an experienced operator/surgeon?

The paper contains quite a few self-citations that don't always seem appropriate (isn't there previous work on which they are based?). I'll frankly admit that I haven't verified this, but please take with verify that you should sometimes refer to the original publications.

Despite the criticisms, I think the work is interesting and valuable, but it needs some improvements.

Detailed remarks:

Detailed comments are included as comments in the original pdf file.

Author Response

Hello
Please find the attached file.

Thank you

Atena

Reviewer 2 Report

It is unclear why using fuzzy logic for the system. Besides, what motivates the fuzzification rules and decisions employed?

The abstract needs to communicate the reason and objective for the work, which are in section 1.1.

As a non-specialist, I wonder how close to laparoscopy the simulation scenario employed is.

The ITBS figure does not seem to communicate all the methods described in Section 2. For example, where is the ResNet employed?

Many other details and methods as the above mentioned need to be addressed by the authors.

Author Response

Hello

Thank you

Atena

Reviewer 3 Report

The manuscript introduces a method for 3D autonomous surgeon's hand movement assessment. Based on the proposed method, the corresponding system is designed and implemented. The evaluations of the proposed method and the system are also performed and analyzed for readers.

Overall, the problem context, the problem statement, the proposed solution, and the efficacy of the solution are well-described. Also, the manuscript is well-organized. It is believed that readers can get valuable insights through the manuscript.

Some suggestions are listed in the following for the authors to revise the manuscript.

1. It is recommended that the evaluations can be made by the proposed method and by the experts. In order to reduce the misjudgment mentioned in the manuscript, several experts can be invited in the evaluation tasks. Thus, the corresponding metrics (e.g., precision, recall, and so on) of the proposed method can be calculated for readers.

2. Based on the proposed method, the position of surgeon's hand seems to be the major factor for the performance. Is it the one and the only factor? Are elapsed time and hand stability also important? Is it possible to detect the above mentioned factors in the proposed method?

3. The related studies are well introduced in the manuscript. It is suggested that a table can be used to summarize the information for readers. Also, remaining issues, challenges, improvement possibilities of those studies can be described. Thus, readers can get more insights into the contribution of the manuscript.

4. The SSD ResNet50 V1 FPN is used in the proposed method. It is suggested that the authors can elaborate on the reason why the model is selected.

5. Please check the captions of Figure 11 and Figure 12. They are the same.

6. Please proofread the manuscript again. Some typos exist in the manuscript (e.g., line 19, line 216 (space), line 371 (space), and so on).

Author Response

Hello

Thank you

Atena

Round 2

Reviewer 1 Report

The article has been significantly improved, unfortunately I am not able to provide detailed corrections, but the authors' explanations seem quite comprehensive.

Author Response

Hello

Thank you for your positive comments.

Reviewer 2 Report

The authors improved the manuscript to address missing details. My main concern is knowing how this paper evolves over the previous other 5 works by the authors. This should be clearly stated in the Introduction.  

Author Response

Hello

One of the main objectives of this study is to assess surgeons’ hand movements in a 3D space, in an autonomous fashion. Although in our previous works, we had improved autonomous skill assessment systems for the peg transfer and tooltip tracking tasks, they were not implemented in a 3D space environment. In previous works, we used only one camera or three cameras, working separately, hence, we were able to track in 2D space. In this study, the autonomous assessment system can detect the laparoscopic instruments and track the surgeon’s hand movements during the experiment by applying a multi-threaded video processing method which has the ability of 3D supervision and evaluation.
